# Anatomical and Transcriptome Analyses of Moso Bamboo Culm Neck Growth: Unveiling Key Insights

**DOI:** 10.3390/plants12193478

**Published:** 2023-10-04

**Authors:** Lin Guo, Tianguo Chen, Xue Chu, Kai Sun, Fen Yu, Feng Que, Zishan Ahmad, Qiang Wei, Muthusamy Ramakrishnan

**Affiliations:** 1State Key Laboratory of Tree Genetics and Breeding, Co-Innovation Center for Sustainable Forestry in Southern China, Bamboo Research Institute, Key Laboratory of National Forestry and Grassland Administration on Subtropical Forest Biodiversity Conservation, School of Life Sciences, Nanjing Forestry University, Nanjing 210037, China; guolin0915@njfu.edu.cn (L.G.); chuxue3722@163.com (X.C.); 15249986398@163.com (K.S.); quefeng@njfu.edu.cn (F.Q.); ahmad.lycos@gmail.com (Z.A.); 2Changzhou Agricultural Technology Extension Center, Changzhou 213000, China; tianguochen2023@163.com; 3Jiangxi Provincial Key Laboratory for Bamboo Germplasm Resources and Utilization, Jiangxi Agriculture University, Nanchang 330045, China; yufen930@163.com

**Keywords:** Moso bamboo, culm neck growth, anatomical and transcriptome analyses, primary thickening growth, lignin synthesis, nutrient transport, secondary metabolism, MYB

## Abstract

The Moso bamboo culm neck, connected with the rhizome and the shoot bud, is an important hub for connecting and transporting the aboveground and belowground systems of bamboo for the shoot bud development and rapid growth. Our previous study revealed that the culm neck generally undergoes six different developmental stages (CNS1–CNS6), according to the primary thickening growth of the underground shoot bud. However, the molecular mechanism of the culm neck development remains unknown. The present study focused on the developmental process of the CNS3–CNS5 stages, representing the early, middle, and late elongation stages, respectively. These stages are densely packed with vascular tissues and consist of epidermis, hypodermis, cortex, and ground tissue. Unlike the hollow structure of the culms, the culm necks are solid structures. As the culm neck continues to grow, the lignin deposition increases noticeably, contributing to its progressive strengthening. For the transcriptome analysis, a total of 161,160 transcripts with an average length of 2373 were obtained from these stages using both PacBio and Illumina sequencing. A total of 92.2% of the reads mapped to the Moso bamboo reference genome. Further analysis identified a total of 5524 novel genes and revealed a dynamic transcriptome. Secondary-metabolism- and transport-related genes were upregulated particularly with the growth of the culm neck. Further analysis revealed the molecular processes of lignin accumulation in the culm neck, which include differentially expressed genes (DEGs) related to cell wall loosening and remodeling and secondary metabolism. Moreover, the upregulations of transcription factors such as *MYBH* and *RSM* in the MYB family play crucial roles during critical transitions in the culm neck development, such as changes in the angle between the rhizome and the culm neck. Our new findings provide essential insights into the cellular roadmaps, transcriptional networks, and key genes involved in the culm neck development.

## 1. Introduction

Moso bamboo (*Phyllostachys edulis*) is one of the fastest growing plants [1] and the most used bamboo species in the bamboo industry [2]. Due to its high economic and ecological value, Moso bamboo is widely planted in East and Southeast Asia, covering approximately 4.43 million hectares in China, representing 84.02% of the global Moso bamboo forest area [3]. Moso bamboo spreads through both asexual and sexual reproduction (monocarpic nature). Asexual reproduction is the most common mode [2]. This occurs when bamboo shoot buds first develop from the rhizome and then grow rapidly in early spring and reach a height of ~20 m within two months. The rhizome is thick and segmented, spreads underground from the mother plant, and can have up to several underground shoot buds at each node [4], although the rhizome is not directly connected to these buds. However, the Moso bamboo culm neck, located at the bottom of the underground shoot bud, connects the shoot bud and the rhizome, supporting the vertical growth of the shoot bud towards above the ground [5]. Thus, the culm neck plays a crucial role in connecting and transporting the aboveground and belowground systems of bamboo for the culm growth and development.

The culm development in Moso bamboo usually undergoes six different developmental stages. First, it begins with the budding stage, during which the shoot bud forms from the rhizome. The second stage is the bud dormancy, when the bud is inactive for a period of time. The third stage is the primary thickening growth, when the bud becomes an active and mature shoot with distinguishable internodes and nodes. At this stage, the pith cavity formation also occurs simultaneously, which plays an important role in determining the different sizes of the internodes and promotes the primary thickening growth [6,7]. The fourth stage is the fast growth stage, during which the mature shoot grows rapidly and reaches a height of ~20 m within two months [1,8,9,10]. The fifth stage is the secondary cell wall thickening process, in which the secondary cell wall formation and lignification occur simultaneously with the rapid growth stage [1]. The final stage is the senescence stage, during which the culm development completes its life cycle. Therefore, understanding the transition between these stages is critical to the rapid growth development.

Several studies have reported that the shoot growth is controlled at multiple levels, such as cell division, rapid division, rapid elongation, and secondary cell wall thickening. Hormones, such as auxin and gibberellin, and their possible downstream target genes, are closely associated with the shoot growth; signal transduction, DNA synthesis, and RNA transcription are the key regulators of the rapid growth [8,10,11,12,13,14]. Our recent study [1] identified internode 18 as a representative internode for the rapid growth, and the culms can grow 114.5 cm/day. This internode includes a 2 cm cell division zone, cell elongation zones up to 12 cm, and a secondary cell wall thickening zone (lignification). The 18-internode growth model includes 14 developmental stages representing critical transition points in the cell division, cell elongation, and secondary cell wall thickening. However, despite these developments, the transition mechanism regulating the rapid growth, particularly in relation to lignin biosynthesis, remains unknown. Lignin deposition in the secondary cell wall is a dynamic process [15], essential for providing structural support, rigidity, and resilience to bamboo [16]. Lignin biosynthesis involves a series of enzymatic reactions that ultimately deposit lignin precursors in the secondary cell wall. This process is regulated by transcription factors (TFs), such as the MYB family, which have emerged as key players in lignin biosynthesis (e.g., the MYB family members, such as *MYB15*, *MYB46*, *MYB58*, *MYB63*, *CCoAOMT*, etc.) [17]. This suggests that the interplay between lignin biosynthesis and MYB TFs represents a crucial nexus in Moso bamboo development during rapid growth [1].

In addition to the shoot developmental stages, our previous study [5] on the culm neck found that, based on the morphological and anatomical analyses, the culm neck is a critical connecting part for nutrient transport and supporting the shoot bud development, and it also undergoes six different developmental stages: CNS1–CNS6 (culm neck stages), according to the primary thickening growth of the shoot bud [7]. The elongation of the culm neck enables the shoot bud to break through the soil and emerge above the ground. As the culm neck emerges, the angle between the culm neck and the rhizome changes, pushing the shoot bud away from the rhizome, ensuring sufficient space for the shoot bud development and its upward movement towards the surface. Therefore, understanding the culm neck development is also very essential for studying the primary thickening growth and the fast growth development. However, the molecular mechanism of the culm neck development remains unknown, despite several developments in the fast growth. Therefore, based on our previous work [5], the present study aimed to understand the developmental process of the CNS3–CNS5 stages (Figure 1a), representing the early (CNS3), middle (CNS4), and late (CNS5) elongation stages, respectively. Both PacBio and Illumina sequencing platforms were used for the transcriptome analysis. The study analyzed gene expressions, including novel genes, in the CNS3–CNS5 stages, and identified significant molecular events associated with the culm neck development. The present study is the first to investigate the culm neck at the molecular level, and it provides novel insights into key genes involved in culm neck growth and development.

## 2. Results

### 2.1. Anatomical Observation of Moso Bamboo Culm Neck

Our previous work reported that the culm necks at the CNS3, CNS4, and CNS5 stages primarily undergo elongation, with the CNS3 stage being the most active growth stage [5]. To investigate the basic biological properties, anatomical structure, and molecular basis of the culm neck at these developmental stages, the present study specifically selected only these stages (CNS3, CNS4, and CNS5) for further detailed analysis. Our results revealed that the culm neck connects the shoot bud and the rhizome. Once the shoot bud matures, the external shape of the culm neck remains consistent or does not change. Notably, the culm neck was completely differentiated in the CNS3 stage, and, subsequently, while maintaining the same differentiation, the culm neck underwent elongation and became longer and larger in the CNS4 or CNS5 stages (Figure 1a). About two-thirds of the bottom of the culm neck is at an angle of about 45 degrees with the shoot bud, while one-third of the top of the culm neck forms an angle of more than 45 degrees with the shoot bud, as described in our previous study [5]. Interestingly, the angle between the rhizome and the culm neck further increased in the CNS5 stage (Figure 1a). Anatomical observations revealed that the culm neck is a solid structure, in contrast to the hollow structure of the internode between the culm nodes. The cross section of the culm neck shows that these three stages were filled with numerous vascular bundles (Figure 1b), distributed from the outside to the inside, epidermis, hypodermis, cortex, and ground tissues (Figure 1c). Furthermore, as the culm neck matures, the level of lignification gradually increases, which is indicated by the deeper red coloration upon staining with saffron dye solution (Figure 1c). The bud scales (a kind of sheath that wraps the neck) attached to the culm neck are similar to bamboo culm sheaths, and the average number of bud scales attached to the culm neck was about 14, as described in our previous study [5].

### 2.2. Culm Neck Transcriptome Sequencing via PacBio and Illumina Platform

The culm necks in the CNS3, CNS4, and CNS5 stages, each with five pooled samples, were sequenced using the PacBio Sequel platform to obtain the full-length transcript sequences. A total of 15,229,362 subreads (16.68 Gb) were obtained, with an average read length of 1096 bp and N50 of 2137 bp (Figure 2a; Appendix A). In addition, 441,914 circular consensus sequences (CCSs) with an average length of 2000 bp were successfully assembled (Appendix A). Further sequence analysis of the sequenced data identified 291,567 full-length reads containing both 5′ and 3′ regions and the poly(A) tail. In addition, 249,071 full-length reads were identified as non-chimeric (FLNC) reads, with an average length of 2444 bp (Appendix A). The FLNC reads with similar sequences were then clustered together using the iterative isoform-clustering (ICE) algorithm. The non-full-length reads were used to correct the consistent sequences in each cluster via Arrow software. A total of 161,160 polished full-length consensus transcripts were generated (Appendix A). The average length of the polished consensus transcripts was 2373 bp, with an N50 of 3510 bp (Appendix A). All the polished consensus isoforms were finally corrected using the NGS reads with LoRDEC software, resulting in 161,160 corrected isoforms with an average N50 length of 3507 bp and an average read length of 2372 bp (Appendix A).

### 2.3. Culm Neck Transcriptome Analysis

The study compared all the corrected transcripts against the Moso bamboo reference genome [18] using GMAP [19]. A total of 148,595 reads (92.2%) were mapped to the reference genome (Appendix A). Among these mapped reads, 76,092 (47.22%) were mapped to the positive strand of the genome, whereas 55,257 (34.29%) were mapped to the opposite strand of the genome. Furthermore, 17,246 reads (10.7%) were mapped to multiple locations, while 12,565 reads (7.80%) remained unmapped to the reference genome (Appendix A). Further analysis revealed distinct patterns of isoform identification. A total of 8676 (11.86%) isoforms were found to match those in the reference genome, while 56,213 (76.83%) novel isoforms were identified from the known genes. In addition, a total of 8278 novel isoforms (11.31%) were identified from the novel genes (Figure 2c; Appendix A). Within these mapped isoforms, it was observed that 43.99% of the genes contained only one isoform, while 56.01% of the genes exhibited more than two isoforms. Notably, approximately 4.2% of the genes had more than 10 isoforms (Figure 2b). Overall, a comprehensive exploration of the Iso-Seq reads revealed that 14,744 genes undergo alternative splicing (AS) events, encompassing about 61.33% of the 24,040 annotated genes. Notably, the predominant AS event (22.75%) was intron retention (RI) (Figure 2d).

Further analysis identified a total of 32,394 lncRNAs divided into four groups: 10,867 (22%) long intergenic non-coding RNAs (lincRNAs), 1676 (20%) intronic sense RNAs, 947 (11%) intronic antisense RNAs, and 3980 sense-overlapping lncRNAs (47%). These classifications were determined using the CNCI, Pfam, PLEK, and CPC methods (Figure 2e). Moreover, our analysis revealed that 6468 genes contained only one poly(A) site (Figure 2f), while 4754 genes exhibited alternative polyadenylation (APA) sites (Figure 2f).

In-depth gene expression analysis revealed that totals of 30,338, 30,841, and 29,267 genes were expressed in the CNS3, CNS4, and CNS5 stages, respectively (Figure 3a). Venn diagram analysis showed that 799, 1223, and 478 genes were expressed specifically in the CNS3, CNS4, and CNS5 stages, respectively (Figure 3a). Gene enrichment analysis using MapMan classified these genes into 35 different functional categories, including protein, RNA, signal transduction, MISC, transport, stress, cell, development, DNA, hormone, secondary metabolism, lipid metabolism, cell wall, and more (Figure 3b). A total of 12,636 genes in the Moso bamboo genome remained unannotated (Figure 3b). The study also analyzed the expressions of TFs and found that the SNF2 family of TFs was the most abundant (Figure 3c). In addition, several TF families, including B3-ARF, C3H, and others, each had more than 200 representative numbers (Figure 3c).

### 2.4. Identification and Annotation of Novel Genes

After the genome mapping, a total of 5524 novel genes were identified, and each of these genes was subjected to annotation via comparison against several databases, such as the GO, SWISS-PROT, NR, NT, COG, KEGG, and Pfam databases (Figure 4b). Among these, 2189 novel genes were expressed in the CNS3, CNS4, and CNS5 stages, while 51, 102, and 34 novel genes exhibited specific expressions in each respective stage, respectively (Figure 4a). Further analysis revealed that some of these novel genes were lignin synthesis-related genes, such as *PAL*, *C4H*, *C3H*, *CCoAOMT*, *F5H*, and *CAD*. In addition, there were cellulose-related novel genes, encompassing precursors, cellulose synthesis, hemicellulose synthesis, cell wall proteins, and genes involved in degradation and modification (Figure 4c,d). Furthermore, a significant number of novel genes were related to various forms of nutrient transporters, including MISC, ABC transporters, metal transporters, and P-and V-ATPases, among others (Figure 4e).

### 2.5. Analysis of Expressed Genes Specifically in the Culm Neck

To further identify expressed genes in the culm neck, the study first analyzed the CNS3, CNS4, and CNS5 stages along with the previously reported genes in the rapidly growing internodes (RGIns) [1] and leaves [20] of Moso bamboo (Figure 5). A total of 22,117 genes were expressed across these tissues, with 117, 175, and 95 genes exclusively expressed in the CNS3, CNS4, and CNS5 stages, respectively (Figure 5). Approximately 65.7% and 76.5% of the genes involved in the CNS3, CNS4, and CNS5 stages were also found to be involved in the fast-growing internodes and leaves, respectively (Figure 5). Gene annotation analysis using MapMan revealed that these genes, which were specifically expressed in the culm neck, were classified into 21 categories, displaying dynamic changes during these developmental stages (Figure 6). Further analysis uncovered interesting genes specifically expressed in the culm neck at different stages, such as genes related to the cell wall, secondary metabolism, and transport (Figure 7). These genes belong to different subcategories, and most of them occurred independently in the different stages of the culm neck (Figure 7).

### 2.6. Analysis of Differentially Expressed Genes (DEGs) in the Culm Neck

To identify differentially expressed genes (DEGs) within the culm neck, the study analyzed the differential expressions across these developmental stages via comparisons. The results showed that 63, 470, and 574 genes were differentially expressed in the CNS4/CNS3, CNS5/CNS3, and CNS5/CNS4 stages, respectively (Figure 8a,b). Among these DEGs, 306 were upregulated, while 637 were downregulated (Figure 8a,b). The MapMan analysis, furthermore, classified these DEGs into 14 different functional categories. The upregulated DEGs encompassed several categories, including amino acid metabolism, auxin signaling, MYB and WRKY TFs, protein synthesis, cellular signaling, sugar transport, and amino acid transport (Figure 8c). Conversely, the downregulated DEGs included major functional classes such as major and minor CHO, cell wall dynamics, gibberellin-related processes, and MISC transport (Figure 8c).

Further analysis revealed different expression patterns within the different functional categories of DEGs. In particular, a substantial number of DEGs related to the cell wall exhibited significant downregulation during the CNS4 or CNS5 stages. These included genes such as *INV* (Invertase), *PMEPCRA* (Pectin Methylesterase), *PE* (PE protein), *XTH9* (Xyloglucan Endotransglucosylase/Hydrolase 9), *PNP* (Polynucleotide Phosphorylase), *EXPAs* (Alpha Expansin), *TCH4* (Touch 4), *XTRs* (Xyloglucan Endo-Transglycosylase-Related), *PGIP2* (Polygalacturonase Inhibiting Protein 2), *GH5* (Glycoside Hydrolase), *GDSL* (Gly-Asp-Ser-Leu), *LRX1* (Leucine-Rich Repeat/Extensin 1), *LRR* (Leucine-Rich Repeat), and *FLA2* (Flowering Locus A2) (Figure 9a). In contrast, *COBL1* (COBRA Like 1) was upregulated in both the CNS4 and CNS5 stages. Notably, genes such *GH3* (Glycoside hydrolase 3), *GH9B8* (Glycosyl Hydrolase 9B8), and *CSLD2* (Cellulose Synthase Like D2) showed initial upregulation in the CNS4 stage, followed by subsequent downregulation in the CNS5 stage (Figure 9a).

Among the DEGs associated with secondary metabolism, a subset displayed consistent upregulation with the progression of the culm neck growth. This group included genes such as *UGT73B5* (UDP-Glucosyl Transferase 73B5), *F5H* (Ferulate 5-Hydroxylase), *HCT* (Hydroxycinnamoyl Transferase), *PRR1* (Pinoresinol Reductase 1), *OMT-1* (O-methyltransferases 1), *HAT* (Histone Acetyltransferase), and *PSY* (Phytoene Synthase). In contrast, the expressions of *SDR* (Short-chain dehydrogenase/reductase) and *OMT-2* (O-methyltransferases 2) followed an opposite pattern (Figure 9b). Furthermore, a clear trend also emerged within the transport-related DEGs. Genes such as *AAP* (Amino Acid Permease), *AAP3*, *AAP6*, and *PROT1* (Proline Transporter 1) exhibited continuous upregulation with the culm neck growth. In contrast, *PTR6* (Peptide Transporter 6), *LAX2* (Like Auxin resistant 2), *PILS5* (PIN-Likes 5), *OCT2* (Organic Cation Transporter 2), *ALF5* (Aberrant Lateral root Formation 5), *PLT5* (Polyol Transporter 5), and *STP1* (Sugar Transporter 1) were downregulated in the CNS4 stage. Interestingly, some of the genes downregulated in the CNS4 stage, such as *PTR6*, *ALF5*, and *PLT5*, were inversely upregulated in the CNS5 stage (Figure 9c).

### 2.7. Analysis of Differentially Expressed WRKY and MYB Families of TFs in the Culm Neck

In addition to the upregulated TF families (Figure 8c), significant observations were made with respect to two specific families of TFs in the culm neck. Among the upregulated TFs, members of the WRKY family, including *WRKY18*, *WRKY46*, *WRKY51*, *WRKY55*, and *WRKY70*, exhibited elevated expression levels either in the CNS4 stage, the CNS5 stage, or both (Figure 10a). In contrast, the MYB family members (Figure 8c), such as *MYB70*, *MYB50*, *MYB117*, *MYB86*, *KAN4*, *MYB46*, *MYB61*, *MYR2*, and *MYB63*, displayed downregulation specifically in the CNS5 stage. However, *MYBH*, *MYB20*, *MYBLU*, *MYB48*, *MYB59*, *RSM3* (*Radialis-like SANT/MYB 3*), and *RSM1* exhibited upregulation in the CNS4 and CNS5 stages (Figure 10b).

Interestingly, as the culm neck grows from the CNS4 to CNS5 stages, the underground shoot bud emerges from the ground in an almost vertical orientation. This change is due to a significant increase in the angle between the culm neck and the rhizome, effectively pushing the shoot bud away from the rhizome to create more space (Figure 10c). Similarly, the elongation of the culm neck plays a crucial role in the emergence of the shoot bud. This process appears to rely on the upregulated expression of MYB-related genes, such as *MYBH* and *MYBLU* (Figure 10b). Importantly, it is worth noting that none of these DEGs in the WRKY and MYB families exhibited expression in the CNS3 stage (Figure 10a,b). In addition, the expression patterns of these key genes were further validated through qRT-PCR analysis. No significant differences were found between the transcriptome data and qRT-PCR analysis (Figure 10d).

## 3. Discussion

The connection between the aboveground and belowground systems of a plant is essential for growth, development, reproduction, and adaptation to environmental conditions through the transport of nutrients, water, hormones, and other signaling molecules. A notable example is the dicotyledons, in which the hypocotyl is a significant part of the response to photomorphogenesis, thermomorphogenesis, and gravitropism. Its specific physiological functions make it an ideal model system for studying cell elongation and tropism growth [21,22,23]. However, the situation differs in monocots, in which the mesocotyl plays a key role in pushing buds (cotyledons) upward from deep water or soil during germination for successful seedling establishment [24,25]. However, due to selective breeding practices, the mesocotyl does not usually elongate above the ground, such as in the modern rice cultivars [26,27].

In contrast, clonal propagation does not rely on hypocotyl elongation but uses vegetative structures, such as rhizomes, stems, roots, or leaves. In Moso bamboo, asexual reproduction is the most common method, which occurs through the rhizome system, where the shoot buds develop [4]. The rhizome is indirectly connected to the shoot bud through the culm neck. The culm neck involvement, in connecting and transporting the aboveground and belowground systems for the shoot bud development, makes its development crucial with six different developmental stages (CNS1–CNS6) [5]. However, the molecular mechanism of the culm neck development remains unknown. There is no previous report on the culm neck at the molecular level, except for one of our previous reports at the anatomical level [5].

Therefore, the current study focused on the developmental process of the CNS3–CNS5 stages, representing the early, middle, and late developmental stages, respectively. The study found that the internal structures in the early stage (CNS3) had fully differentiated, with epidermis, hypodermis, cortex, and ground tissues (Figure 1), indicating that culm neck differentiation occurs much earlier than the differentiation of the primary thickening growth of the shoot bud, which is a prerequisite for the primary thickening growth of the shoot bud [7]. The current study found that the culm necks are solid structures with the pseudonodes (Figure 1a), which is different from the hollow structures and the true nodes and internodes in the culm. Furthermore, the degree of lignification gradually increased from the CNS3 to CNS5 stages (Figure 1b,c), implying that the lignin deposition significantly supports the structural integrity of the culm neck. The lignification in the culm neck also occurs much earlier than in the primary thickening growth of the shoot bud [1,7]. These particular anatomical features may play a critical role in providing the necessary support and facilitating nutrient transport pathways for the primary thickening and subsequent rapid growth of the shoot.

To gain a deeper insight into the molecular mechanisms regulating the developmental process of the CNS3–CNS5 stages, the current study performed a transcriptome analysis. Interestingly, the study found 5524 novel genes, accounting for 10% of the Moso bamboo genome, that are involved in specialized cellular processes that fine-tune biological pathways involved in the culm neck development (Figure 4), which has also greatly improved the information about the Moso bamboo genome. To understand the support and nutrient transport functions of the culm neck, the study further focused on the DEGs related to the cell wall, secondary metabolism, and nutrient transport. The DEGs related to cell wall loosening and remodeling, such as *PMEPCRA* [28], *EXPA1* [29], *EXPA4* [30], etc., showed significant downregulation (Figure 9). This indicates a possible restriction of the cell growth, which is likely due to the dynamic lignification process in the culm neck. Conversely, the genes associated with lignin accumulation, such as *F5H* [31], *HCT* [32], *PRR1* [33], were significantly upregulated (Figure 9). This suggests a possible acceleration of the lignification through the regulation of these related genes in the culm neck. The molecular components responsible for the lignin deposition in the culm neck were found to be similar to those found in bamboo culms [1,34]. Notably, a genome-wide analysis pinpointed the involvement of key genes, including *PALs*, *C4Hs*, *C3Hs*, *4CLs*, *HCTs*, *CCRs*, *CCoAOMTs*, *CAD*, *F5Hs*, and *COMT*, in the lignin biosynthesis process in Moso bamboo [35]. Furthermore, considering the role of the culm neck in nutrient transport from the rhizome to the shoot, the study also observed the significant upregulation of the genes related to amino acid transporters, such as *AAP3*, *AAP6* [36,37], and *PROT1* [38]. In particular, *AAP6*, expressed in the xylem parenchyma or phloem in *Arabidopsis* [36,39], has been reported to significantly enhance amino acid absorption in rice roots [40]. These results shed light on the complex molecular basis of the culm neck development and its multiple functions.

In addition, *WRKY18*, a member of the WRKY TF family, was upregulated in the CNS5 stage and plays a crucial role in regulating the rapid responses to sugar metabolism [41]. In *Arabidopsis*, *MYBH*, a member of the MYB TF family, has been reported to be involved in the positive regulation of darkness-induced hypocotyl elongation [42]. Similarly, the upregulation of *MYBH* in the CNS4 and CNS5 stages suggests that it is a molecular component responsible for promoting the elongation of the culm neck in the underground darkness. In addition, as members of the MYB TF family, both *MYB59* and *MYR2* are involved in distinct roles. While *MYB59* is involved in the regulation of nutrient transport [43], *MYR2* has been reported to regulate nitrogen reutilization [44]. Additionally, in *Arabidopsis*, *RSM1*, another member of the MYB TF family, has been reported to negatively regulate the apical hook bending [45]. Interestingly, in the current study, as the members of a small subfamily of single MYB proteins, the upregulation of *RSM1* and *RSM3* may serve as a key factor in the angle change between the rhizome and the culm neck (Figure 10c). The angle change implies that the shoot bud, which is primarily thickened, can obtain more space and grows upward, reaching the ground instead of spreading underground. 

Like leaves affected by negative gravity, the culm neck grows upward against gravity. The leaf angle serves to orient the leaves and other aboveground organs toward the sun and anchor them below ground [46,47]. Several genes are known to regulate leaf angles [46,47]. For instance, the TF *OsBZR1* regulates *OsIAA6* by interacting with the promoter of *OsIAA6*, which, in turn, controls the leaf angles in rice by suppressing the auxin response factor, *OsARF1* [48]. Similarly, *OsARF19* regulates the leaf angle by positively regulating *OsGH3-5* and *OsBRI1* and promoting cell division on the adaxial side of the lamina joint in rice [49]. Furthermore, *OsMYB7*, a member of the MYB TF family, plays a role in determining the leaf angle during specific developmental stages in rice. When *OsMYB7* is overexpressed, it results in the development of wide-angled leaves, while knockout mutants of *osmyb7* display erect leaves. *OsMYB7* achieves this effect by increasing the thickness of sclerenchyma cell walls, primarily by elevating the cellulose content [50]. Interestingly, in the current study, the MYB TF family members such as *RSM1* and *RSM3* play a role in the culm neck angle. These findings suggest that the molecular components responsible for the culm neck angle growth are similar to those found in the leaf angle growth. Furthermore, several members of the WRKY and MYB TFs were not expressed in the CNS3 stage (Figure 10a,b). However, these members were significantly expressed in the CNS4 and CNS5 stages (Figure 10a,b), indicating that these TFs are crucial for the transition of the morphological changes of the culm neck.

## 4. Materials and Methods

### 4.1. Plant Material

Moso bamboo culm necks, located at the bottom of the underground shoot bud, were collected from Jinyuan Forest Farm, Daao Village, Yichun City, Jiangxi Province, China. This region (27°33′–29°06′ N and 113°54′–116°27′ E) has a humid subtropical climate with an average annual precipitation of 1631.5 mm (64 inches). The soil at the collection point primarily consists of red soil and paddy soil. First, the underground shoot buds of Moso bamboo were collected at different developmental stages. Subsequently, the culm necks, used for the anatomical and transcriptome analyses, were carefully separated from the shoot buds. The developmental stages of the shoot buds of Moso bamboo were described according to our previous report [7]. The culm neck stages (CNSs) CNS3, CNS4, and CNS5, representing the early (CNS3), middle (CNS4), and late (CNS5) elongation stages, respectively, were described in accordance with our previous report [5]. Each developmental stage of the culm neck corresponds to a specific developmental stage of the shoot bud. The culm neck stages are depicted in Figure 1a.

### 4.2. Light Microscope Observation

Approximately 0.5 cm^3^ of the young fresh tissues of Moso bamboo culm necks at different developmental stages were collected and then fixed in formalin–acetic acid–70% alcohol (FAA, *v*/*v*) buffer. The paraffin sectioning and observation were performed and observed under a Leica DM2500 light microscope (Leica, Wetzlar, Germany), as described in our previous report [51].

### 4.3. RNA Extraction and Transcriptome Sequencing

The fresh tissues of the culm necks at different developmental stages (CNS3, CNS4, and CNS5) [5] were collected, and each stage had five biological replicates. The collected samples were immediately frozen in liquid nitrogen and stored at −80 °C. Total RNA extraction and quality checks were performed following the methods described in our previous reports [9,52]. Briefly, the total RNA was extracted using the RNAprep Pure Kit (DP441) (TIANGEN Biotechnology, Beijing, China). The RNA ratio (OD260/OD280) was determined using a NanoDrop 1000 spectrophotometer (Thermo Scientific, Waltham, MA, USA). The quality of the RNA was also confirmed through agarose gel electrophoresis. The RNA integrity number (RIN) values of the RNA samples were further checked using an Agilent Bioanalyzer 2100 system (Agilent Technologies, Palo Alto, CA, USA). The samples with sharp RNA bands on the agarose gel electrophoresis and RIN values > 8.0 were selected for RNA-sequencing library preparation. Then, the libraries were prepared by fragmenting the cDNA of the RNA samples for Illumina sequencing, which was performed at Novogene Biotechnology Company (Beijing, China).

In order to obtain full-length transcripts, full-length transcriptome sequencing was also performed in the present study. Already quality-passed RNA samples were further selected, with each stage having five biological replicates. Subsequently, the samples were carefully pooled together in equal quantities to create one sample for PacBio SMRT sequencing. The full-length cDNAs were synthesized using the SMRTer PCR cDNA Synthesis Kit (Clontech, Mountain View, CA, USA). A fraction containing fragments > 4 kb was enriched and collected, using the BluePippin Size-Selection System (Sage Science, Beverly, MA, USA). Finally, the enriched fraction was mixed equally (equimolar) with the unscreened fractions to create a mixed SMRT cell that was run on the PacBio Sequel platform. The experiment was performed at Novogene Biotechnology Company (Beijing, China). 

### 4.4. Transcriptome Sequence Processing and Analysis

The raw reads from Illumina sequencing were first checked for quality using FastQC [53], and the adaptors and low-quality sequences with Qphred scores ≤ 20 or N ratios more than 10% were removed from the raw reads. Then, the clean reads were used for the de novo assembly along with the PacBio sequencing data. The raw PacBio subreads were filtered and subjected to circular consensus (CCS) using SMRTlink 4.0 (Pacific Biosciences, Menlo Park, CA, USA). To enhance the accuracy, the clean reads from Illumina sequencing were used to correct the polished consensus reads using LoRDEC software with default parameters [54,55]. The redundancy in the corrected contigs was removed with the sequence identity cut-off set to 95% using CD-HIT v 4.7 [56].

After transcriptome assembly, the transcripts were finally annotated by comparing their sequences against several databases, such as Gene Ontology (GO) [57], SWISS-PROT (protein sequence) [58], the NCBI non-redundant protein database (NR) [59], the NCBI non-redundant nucleotide database (NT), Pfam (protein family) [60], Clusters of Orthologous Groups of Proteins (KOG/COG) [61], and the Kyoto Encyclopedia of Genes and Genomes Ortholog Database (KEGG) [62], using ncbi-blast-2.7.1+, Diamond v0.8.36 [63] or the HMMER 3.1 package [64].

Transcription factors (TFs) were annotated using the iTAK program [65]. The non-redundant transcripts were further analyzed to predict non-coding transcripts through the PLEK (predictor of long non-coding RNAs and messenger RNAs based on an improved *k*-mer scheme) [66] and CNCI (Coding–Non-Coding Index) [67]. The resulting transcripts were then analyzed using the Coding Potential Calculator (CPC) [68] and Pfam [69] to finally identify long non-coding RNAs (LncRNAs).

For the differential gene expression analysis, the clean reads from Illumina sequencing were aligned to the non-redundant consensus transcript using Bowtie2 [70]. After the alignments, the raw read counts for each transcript were obtained using RSEM [71], and the read counts were further normalized to FPKM (fragments per kilobase exon per million mapped fragments). The read counts were subjected to the DESeq2 package [72] to identify differentially expressed genes (DEGs). The genes with adjusted *p*-values < 0.05 and absolute fold changes no less than 2 were identified as DEGs. The expression patterns of the DEGs were visualized using MapMan (v.3.5.1R2) [73]. In addition, the mapping file of the assembled transcripts was generated using Mercator [74].

### 4.5. Quantitative Real-Time Polymerase Chain Reaction Analysis

To verify the transcriptome results, 4 genes for the culm neck were randomly selected, and gene-specific primers were designed using Primer Premier 5 software (http://www.premierbiosoft.com/ (accessed on 24 September 2023)); their sequences are listed in Appendix A. Total RNA was extracted from the culm necks at three different developmental stages (CNS3, CNS4, and CNS5), as mentioned above. The first strand of cDNA was synthesized using the TransScript One-Step gDNA Removal and cDNA Synthesis SuperMix Kit (Transgene). Quantitative real-time PCR (qRT-PCR) was performed on an ABI StepOne Plus Real-Time PCR System (Applied Biosystems, Waltham, MA, USA) using the ChamQTM Universal SYBR quantitative real-time polymerase chain reaction Master Mix Kit (Vazyme, Nanjing, China), according to the manufacturer’s instructions. To ensure reliable normalization, the tonoplast intrinsic protein 41 (*TIP41*) gene was used as an internal control [75]. The relative abundance of each gene was calculated from the 2^ΔCq^ values between the target gene and the reference gene, with three replicates for each gene [76].

### 4.6. Statistical Analysis and Replicates

To ensure the reliability and robustness of our results, each stage of culm neck development (CNS3, CNS4, and CNS5) was represented by three biological replicates for anatomical analysis and five biological replicates for transcriptome sequencing. For qRT-PCR, three replicates were performed for each gene, providing additional validation of the transcriptome data. The data obtained from these replicates were subjected to appropriate statistical tests to determine the significance of the differences in the gene expression levels.

## 5. Conclusions

In conclusion, our study focused on the anatomical and transcriptome analyses of the CNS3, CNS4, and CNS5 stages. The culm neck is a solid structure and differentiates in the CNS3 stage and further elongates in the CNS4 or CNS5 stages. The increased angle between the rhizome and the culm neck in the CNS5 stage facilitates the upward growth of the shoot bud (Figure 11). The lignification gradually increases as the culm neck matures. Transcriptome analysis identified DEGs in these stages, along with lncRNAs divided into four distinct groups. Moreover, we identified distinct patterns of isoforms, including novel isoforms, and found that 14,744 genes undergo AS events, with intron retention (RI) predominant. Gene enrichment analysis classified the DEGs into different functional categories, including signal transduction, hormone, secondary metabolism, lipid metabolism, cell wall, etc. 

Further we identified a total of 5524 novel genes related to the cell wall, secondary metabolism, and nutrient transport. Among these, 51, 102, and 34 novel genes exhibited specific expressions in the CNS3, CNS4, and CNS5 stages, respectively. These novel genes are related to lignin synthesis, cellulose synthesis, hemicellulose synthesis, nutrient transporters, etc. In addition, the genes involved in these stages are also involved in the fast-growing internodes and leaves. The members of WRKY and MYB TFs exhibited elevated expression levels, particularly during the transition from the CNS4 to CNS5 stages. The upregulation of *MYBH* and *MYBLU*, both members of the MYB TF family, in the CNS4 and CNS5 stages, contributes to the elongation of the culm neck, leading to a significant increase in the angle between the culm neck and the rhizome. The changing angle between the culm neck and the rhizome, particularly marked in the CNS5 stage, enables the emergence of the shoot bud in an upward direction. 

Overall, our study revealed that the WRKY and MYB families play distinct roles in the elongation of the culm neck and provide valuable insights into the complex processes controlling the shoot bud emergence and the culm neck elongation (Figure 11). The interplay between the TFs and the dynamic expression patterns requires further studies to understand the transition mechanisms between these critical developmental events. 

## Figures and Tables

**Figure 1 plants-12-03478-f001:**
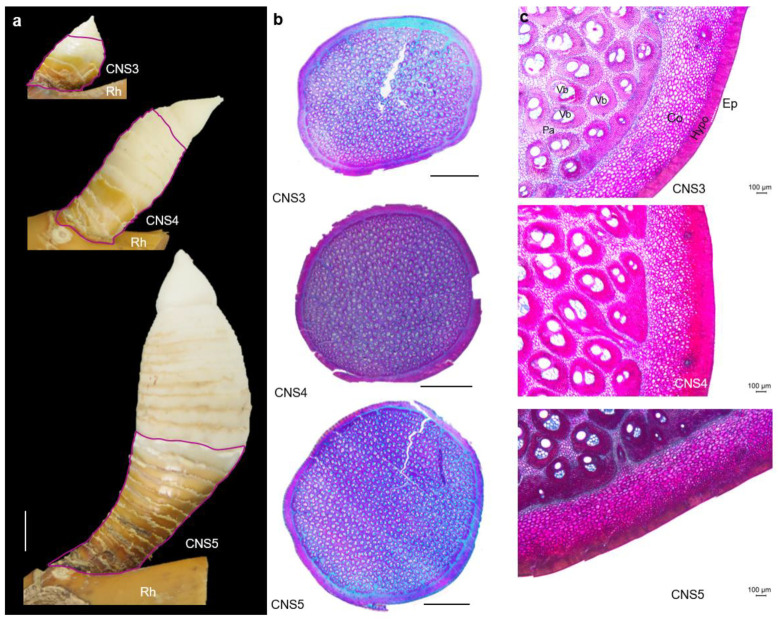
The (**a**) morphological and (**b**,**c**) anatomical characterization of Moso bamboo culm neck. (**a**) The closed purple lines show the morphology of the culm neck after removing the bud scales, illustrating the angle between the culm neck and the shoot bud connected to the rhizome (Rh). The culm neck stages CNS3, CNS4, and CNS5 represent the early, middle, and late elongation stages, respectively. (**b**) The cross section of the culm neck with a solid structure. (**c**) The cross section shows the tissue types and cell layers from the outside to the inside: the white stars indicate the epidermis (Ep); Hypo, hypodermis; Co, cortex; Vb, vascular bundle; Pa, parenchyma cell. The gradual lignification is indicated by the deeper red coloration upon staining with saffron dye solution. Scale bars in (**a**–**c**) = 1 cm, 0.25 cm, and 100 µm, respectively.

**Figure 2 plants-12-03478-f002:**
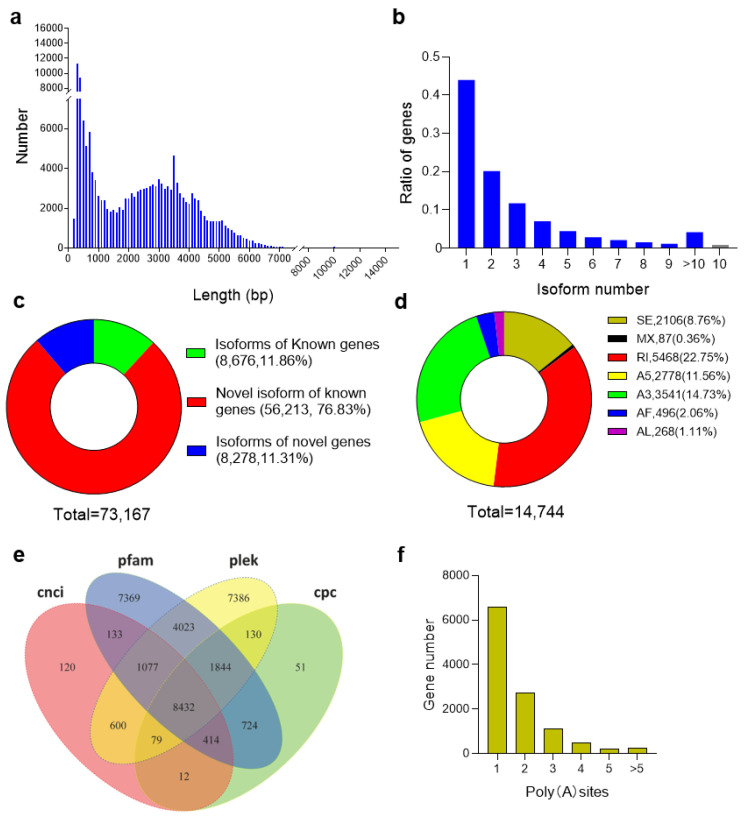
The transcriptome profile of Moso bamboo culm neck. (**a**) Distribution of transcript length. (**b**) Total number of isoforms and their proportions of total genes. (**c**) Proportions of isoforms of the known genes, novel isoforms of the known genes, and isoforms of novel genes in total transcripts. (**d**) Predicted alternative splicing (AS) events from the Iso-Seq reads. (**e**) Total long non-coding RNA (lncRNA) identified by CNCI (Coding–Non-Coding Index), Pfam, PLEK (predictor of long non-coding RNAs and messenger RNAs based on an improved k-mer scheme), and CPC (Coding Potential Calculator). (**f**) Numbers of poly(A) sites in total genes.

**Figure 3 plants-12-03478-f003:**
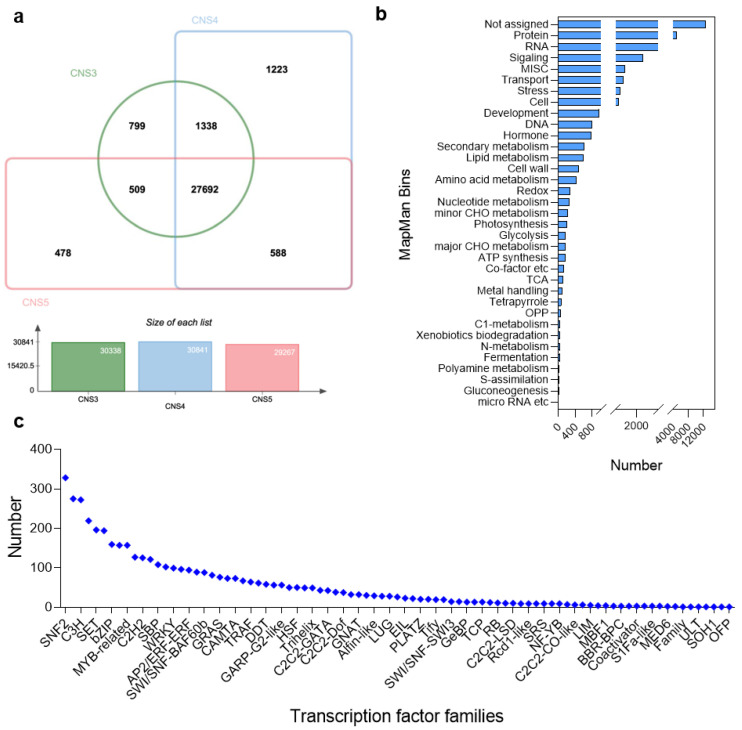
Overview of the total number of genes expressed in Moso bamboo culm neck. (**a**) A Venn diagram shows the number of genes expressed in the culm neck stages CNS3 (early), CNS4 (middle), and CNS5 (late). (**b**) Enrichment analysis of the expressed genes by MapMan 3.6.0RC1 software. (**c**) Transcription factor families and the total number of members in each family expressed in the culm neck.

**Figure 4 plants-12-03478-f004:**
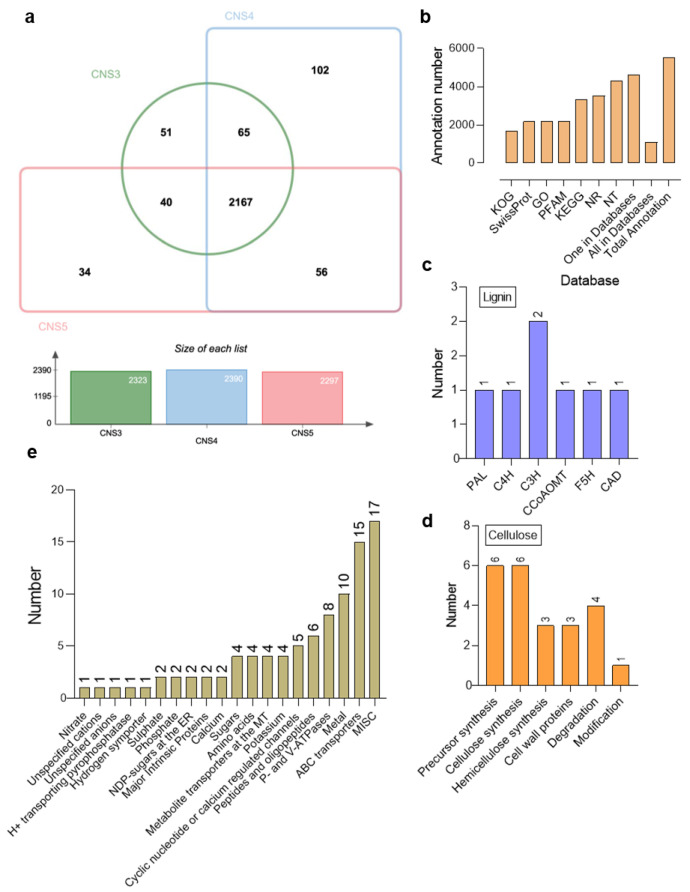
Details of the novel genes expressed in Moso bamboo culm neck. (**a**) A Venn diagram shows the number of novel genes expressed in the culm neck stages, such as CNS3 (early), CNS4 (middle), and CNS5 (late). (**b**) The total of number of novel genes annotated using different databases, such as KOG, SWISS-PROT, GO, Pfam, KEGG, NR, and NT. (**c**) The number of novel genes related to (**c**) lignin, (**d**) cellulose, and (**e**) transporters.

**Figure 5 plants-12-03478-f005:**
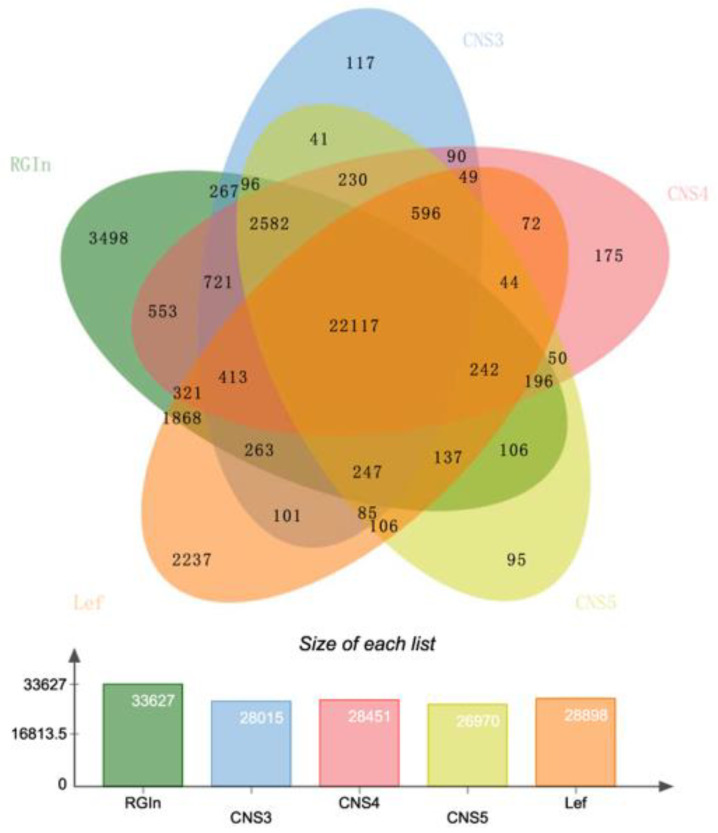
A Venn diagram shows the total number of genes expressed in rapidly growing internodes (RGIns) [1] and leaves [20], and the culm neck stages CNS3 (early), CNS4 (middle), and CNS5 (late).

**Figure 6 plants-12-03478-f006:**
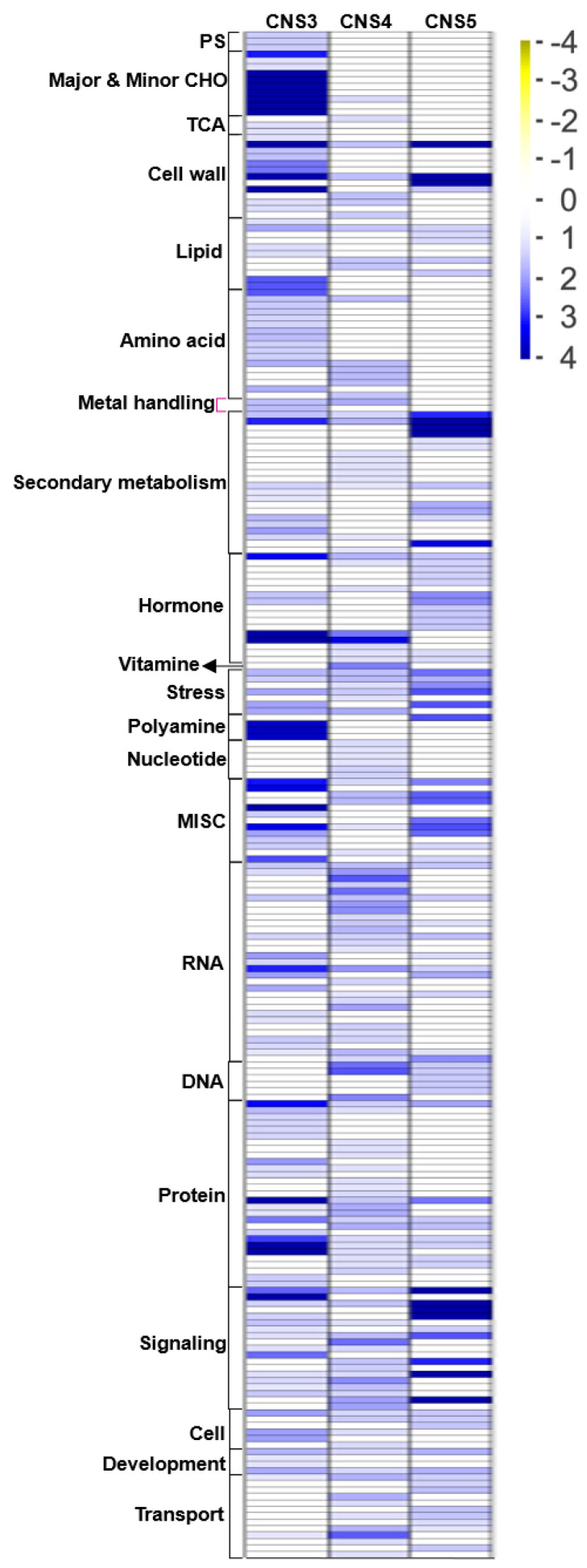
Overview of annotation of genes specifically expressed in the culm neck stages CNS3 (early), CNS4 (middle), and CNS5 (late) by MapMan 3.6.0RC1 software. The color scale represents the log2 (FPKM).

**Figure 7 plants-12-03478-f007:**
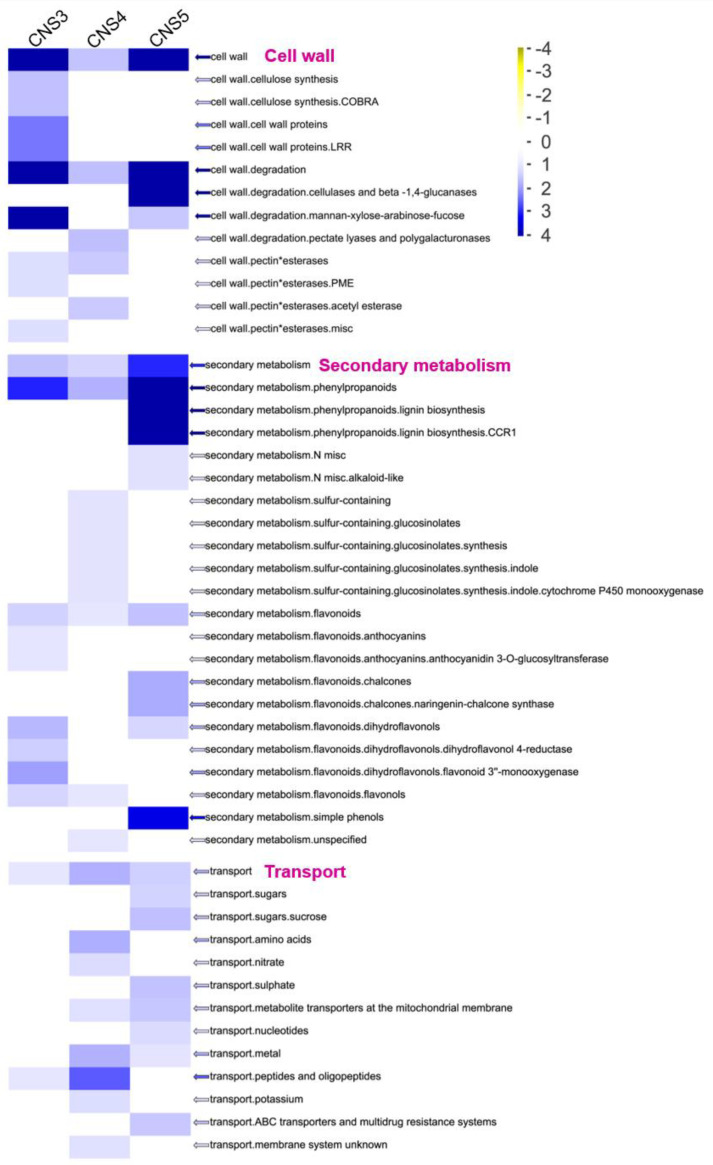
Genes specifically expressed in the culm neck stages CNS3 (early), CNS4 (middle), and CNS5 (late) related to cell wall, secondary-metabolism, and transport genes. The color scale represents the log2 (FPKM).

**Figure 8 plants-12-03478-f008:**
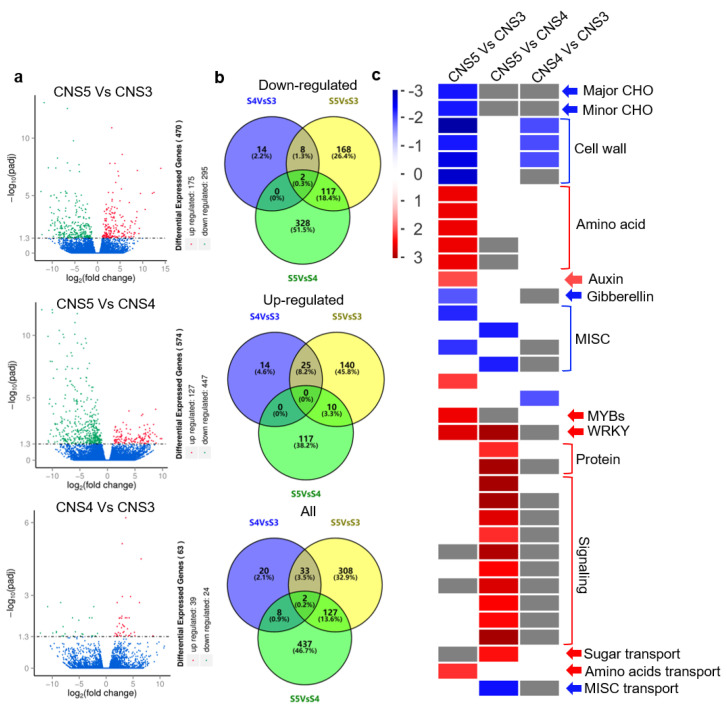
Overview of differentially expressed genes (DEGs) between the CNS3, CNS4, and CNS5 stages of the culm neck. (**a**) Volcano plots and (**b**) Venn diagrams show the different comparisons of DEGs between these stages (CNS5 vs. CNS3, CNS5 vs. CNS4, and CNS4 vs. CNS3). (**c**) Overview of enrichment analysis of DEGs by MapMan 3.6.0RC1 software. The color scale represents the log2-fold change (differential expression) values.

**Figure 9 plants-12-03478-f009:**
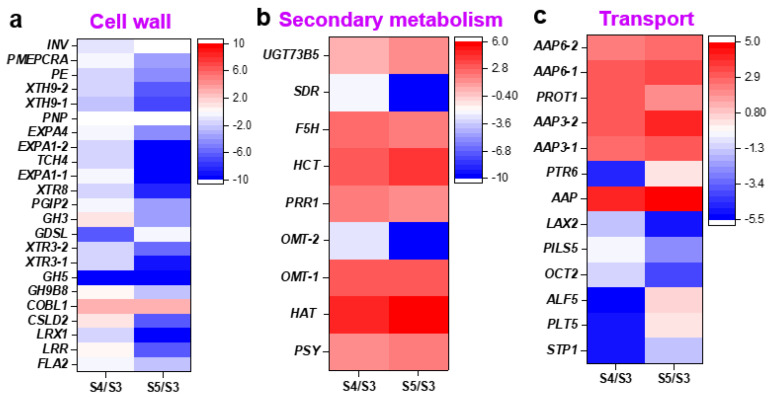
Differentially expressed genes (DEGs) related to cell wall (**a**), secondary-metabolism (**b**), and transport (**c**) in different comparisons of the culm neck stages (CNS5 vs. CNS3, CNS5 vs. CNS4, and CNS4 vs. CNS3). The color scales represent the log2-fold change (differential expression) values.

**Figure 10 plants-12-03478-f010:**
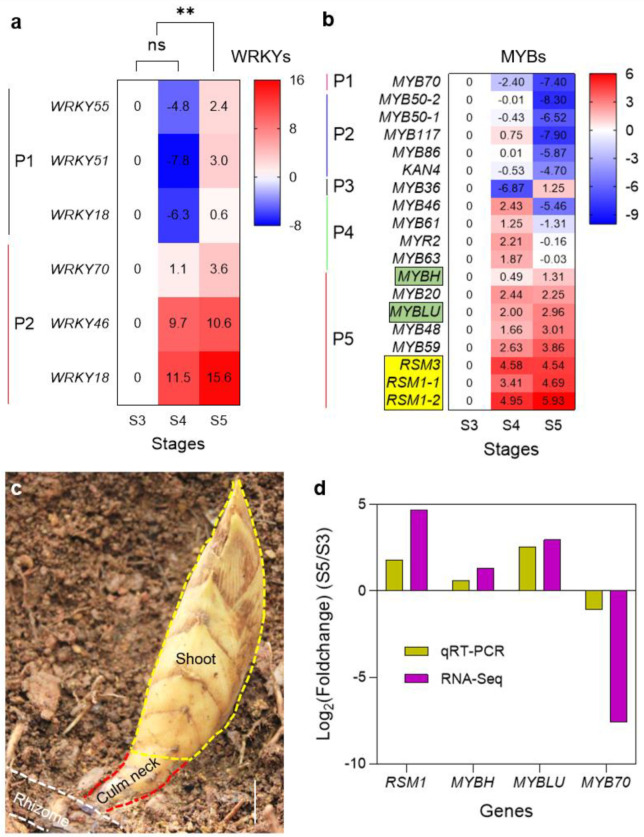
The representative members of the transcription factor families (**a**) WRKY and (**b**) MYB in the culm neck stages CNS3 (early), CNS4 (middle), and CNS5 (late). The color scales represent the log2-fold change (differential expression) values. In (**a**), "ns" and the double asterisk indicate no significance and *p* < 0.01, respectively. (**c**) The morphology of the culm neck and the shoot with sheaths in a natural environment. The culm neck is located between the rhizome and the shoot. The yellow oval represents the upward-growing shoot, and the white lines represent the rhizome. The angle and elongation of the culm neck facilitate the upward growth of the shoot bud. Scale bars = 3 cm. (**d**) Quantitative real-time PCR analysis (qRT-PCR) of the selected genes in MYB-family TFs. Data are means ± SDs (*n* = 3).

**Figure 11 plants-12-03478-f011:**
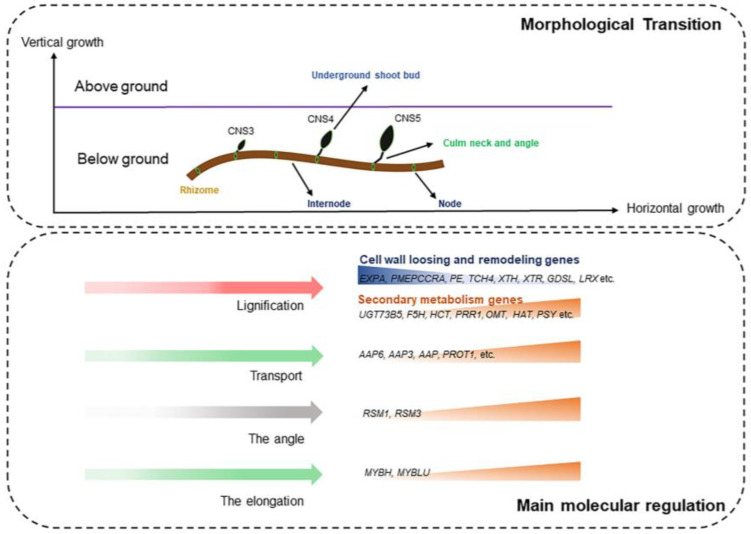
The main results of the anatomical and transcriptomic analyses of Moso bamboo culm neck stages CNS3 (early), CNS4 (middle), and CNS5 (late). It connects with the rhizome and the underground shoot bud, differentiates in the CNS3 stage, and undergoes further elongation in the CNS4 and CNS5 stages. The genes related to cell wall loosening and remodeling and secondary metabolism play a key role in regulating the lignification. The angle between the rhizome and the culm neck, together with the elongation of the culm neck, enables the shoot bud to emerge in an upward direction. Notably, the upregulation of RSM1 and RSM3, as well as MYBH and MYBLU, regulates the angle change and the elongation of the culm neck, respectively, for the upward growth of the shoot bud.

## Data Availability

The data that support the findings of this study are available from the corresponding authors upon reasonable request.

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
