# Peer review of "Anatomical and Transcriptome Analyses of Moso Bamboo Culm Neck Growth: Unveiling Key Insights"

_plants, 2023, doi:10.3390/plants12193478_

Round 1

Reviewer 1 Report

It's great to read a study involving anatomical and transcriptome analyses of Moso bamboo culm neck growth. Here are some minor revision comments to consider:

Abstract: Be specific with abstract words, which provides a concise summary of your study, including the key objectives, methods, results, and conclusions.

Title: The title is clear and informative. However, consider adding a bit more specificity to capture the essence of your research. For example, mention the key findings or the focus of the anatomical and transcriptome analyses. A revised title could be something like "Anatomical and Transcriptome Analyses of Moso Bamboo Culm Neck Growth: Unveiling Key Insights."

Introduction: In the introduction section, elaborate more about adequate background information about Moso bamboo and why studying culm neck growth is important. Additionally, briefly mention the existing knowledge gaps or questions that your research aims to address.

Methods: This should include sample collection, preparation, data collection techniques, and any statistical or bioinformatics tools utilized. Clarity in methods is crucial for reproducibility.

Results: Results are in good format. In the results section, present your findings clearly and concisely.

Figures : Figures 1 to 10 are very comprehensive.

Discussion: Expand on the implications of your findings in the discussion section. Discuss how your results contribute to the existing body of knowledge about Moso bamboo growth and culm neck development. Consider comparing your findings to prior studies and explaining any differences or similarities.

Conclusion: In the conclusion, summarize the main takeaways from your research. Restate the significance of your findings and their potential applications or implications for the field of bamboo cultivation or plant biology.

References: Double-check the references are up-to-date, as some references seem incomplete.

Language and Clarity: Good and clear.

Paper is very good. I would like to recommend its publication after minor revision.

Author Response

Reviewer #1

Response:

We sincerely thank the reviewer for the positive responses to our manuscript and the concise summary of our work. The reviewers’ comments encouraged us to carefully revise and refine our manuscript. We hope that the revised manuscript has been significantly improved and is suitable for publication in Plants.

Reviewer’s comment 1 (Summary):

It's great to read a study involving anatomical and transcriptome analyses of Moso bamboo culm neck growth. Here are some minor revision comments to consider.

Response:

We sincerely thank the reviewer for showing interest and for the positive response to our review, and we appreciate the very positive and constructive summary of our work. The reviewer’s comments encouraged us to carefully revise and refine our manuscript. We hope that the revised manuscript has significantly improved and is suitable for publication in Plants.

Specific comments:

Comment 2:

Abstract: Be specific with abstract words, which provides a concise summary of your study, including the key objectives, methods, results, and conclusions.

Response:

We appreciate the reviewer's feedback on the abstract. We have carefully reviewed the abstract and believe that it effectively summarizes our study with clarity and specificity.

Comment 3:

Title: The title is clear and informative. However, consider adding a bit more specificity to capture the essence of your research. For example, mention the key findings or the focus of the anatomical and transcriptome analyses. A revised title could be something like "Anatomical and Transcriptome Analyses of Moso Bamboo Culm Neck Growth: Unveiling Key Insights."

Response:

We thank the reviewer for this comment and have changed the title, which is now reads: “Anatomical and transcriptome analyses of Moso bamboo culm neck growth: Unveiling key insights”. This is more in line with the focus of this paper.

Comment 4:

Introduction: In the introduction section, elaborate more about adequate background information about Moso bamboo and why studying culm neck growth is important. Additionally, briefly mention the existing knowledge gaps or questions that your research aims to address.

Response:

We thank the reviewer for their feedback on the introduction, in which we have included some additional points about lignin biosynthesis in bamboo and the general role of MYB family transcription factors. We believe this summarizes the background information effectively. It now reads: Please see line numbers 76-86.

“However, despite these developments, the transition mechanism regulating the rapid growth, particularly in relation to lignin biosynthesis, remains unknown. Lignin deposition in the secondary cell wall is a dynamic process [15], essential for providing structural support, rigidity, and resilience to bamboo [16]. Lignin biosynthesis involves a series of enzymatic reactions that ultimately deposit lignin precursors in the secondary cell wall. This process is regulated by transcription factors (TFs), such as the MYB family, which have emerged as key players in lignin biosynthesis, e.g., the MYB family members, such as MYB15, MYB46, MYB58, MYB63, CCoAOMT, etc. [17]. This suggests that the interplay between lignin biosynthesis and MYB TFs represents a crucial nexus in Moso bamboo development during the rapid growth [1].”

The newly added reference is copied below:

  1. Zhu, Y.; Huang, J.; Wang, K.; Wang, B.; Sun, S.; Lin, X.; Song, L.; Wu, A.; Li, H. Characterization of lignin structures in Phyllostachys edulis (Moso bamboo) at different ages. Polymers 2020, 12, 187, doi:10.3390/polym12010187.
  2. Li, L.; Yang, K.; Wang, S.; Lou, Y.; Zhu, C.; Gao, Z. Genome-wide analysis of laccase genes in Moso bamboo highlights PeLAC10 involved in lignin biosynthesis and in response to abiotic stresses. Plant Cell Rep 2020, 39, 751-763, doi:10.1007/s00299-020-02528-w.
  3. Xiao, R.; Zhang, C.; Guo, X.; Li, H.; Lu, H. MYB transcription factors and its regulation in secondary cell wall formation and lignin biosynthesis during xylem development. Int. J. Mol. Sci. 2021, 22, 3560, doi:10.3390/ijms22073560.

Comment 5:

Methods: This should include sample collection, preparation, data collection techniques, and any statistical or bioinformatics tools utilized. Clarity in methods is crucial for reproducibility.

Response:

We thank the reviewer for this comment. We have ensured that the sample collection, preparation, data collection, statistical, and bioinformatics tools utilized have all been included for clarity and reproducibility. Additionally, we have provided a section for statistics and supplementary materials to support our findings. The statistics section now reads: Please see line numbers 519-525.

4.6. Statistical analysis and replicates

To ensure the reliability and robustness of our results, each stage of culm neck development (CNS3, CNS4, and CNS5) was represented by three biological replicates for anatomical analysis and five biological replicates for transcriptome sequencing. For qRT-PCR, three replicates were performed for each gene, providing additional validation of the transcriptome data. The data obtained from these replicates were subjected to appropriate statistical tests to determine the significance of differences in gene expression levels.”

Comment 6:

Results: Results are in good format. In the results section, present your findings clearly and concisely.

Response:

We thank the reviewer for their feedback on the results. We have carefully reviewed the results and believe that it summarizes our findings clearly.

Comment 7:

Figures: Figures 1 to 10 are very comprehensive.

Response:

We thank the reviewer for their feedback on the figures and have ensured that all the figures stand on their own.

Comment 8:

Discussion: Expand on the implications of your findings in the discussion section. Discuss how your results contribute to the existing body of knowledge about Moso bamboo growth and culm neck development. Consider comparing your findings to prior studies and explaining any differences or similarities.

Response:

We thank the reviewer for this comment and have revised the discussion by comparing our results with previous studies. We have also highlighted how the results contribute to the existing knowledge of Moso bamboo growth and culm neck development.

Comment 9:

Conclusion: In the conclusion, summarize the main takeaways from your research. Restate the significance of your findings and their potential applications or implications for the field of bamboo cultivation or plant biology.

Response:

We thank the reviewer for bringing up this valid point and have thoroughly revised the conclusion by restating our results and highlighting their potential applications in the field of bamboo biology.

Comment 10:

References: Double-check the references are up-to-date, as some references seem incomplete.

Response:

We thank the reviewer for this comment. We have thoroughly checked all the references in the revised manuscript and ensured that they are formatted according to the journal's style guidelines.  

Comment 11:

Language and Clarity: Good and clear. Paper is very good. I would like to recommend its publication after minor revision.

Response:

We appreciate the positive feedback and recommendation for publication after minor revisions and have revised the contents of the manuscript as per your suggestions.

Reviewer 2 Report

1. This manuscript is well-prepared and it aims to continue the authors' previous findings and further analysis of the molecular mechanism focusing on the development of Moso bamboo culm neck. The role of some transcription factors was identified by the qRT-PCR and definitely, these results should be further validated by a series of molecular tools in the future.

2. L19: Avoid using first-person writing throughout the manuscript.

3. Keywords: These should be key terms but did not appear in the manuscript title.

4. Introduction: Add some background information about lignin biosynthesis in bamboo and the general roles of transcription factors that were investigated in this study, e.g. MYB family.

5. Figure legends: should be enriched by adding information about abbreviations, e.g. the developmental stages, to make it easy to read figures individually and gene names should be in italics.

6. M&M: Add a section for statistics. Add information about how many replicates were conducted.

7. Discussion: The authors should discuss the general molecular mechanism more in detail on the angle change and comprehensively describe the present findings when compared to the literature.

8. Figure 11: I am curious about the related length of the culm neck when compared with the shoot buds. It seems too long in this figure.

Author Response

Reviewer #2

Response:

We sincerely thank the reviewer for the positive responses to our manuscript and the concise summary of our work. The reviewers’ comments encouraged us to carefully revise and refine our manuscript. We hope that the revised manuscript has been significantly improved and is suitable for publication in Plants.

Reviewer’s comment 1 (Summary):

This manuscript is well-prepared and it aims to continue the authors' previous findings and further analysis of the molecular mechanism focusing on the development of Moso bamboo culm neck. The role of some transcription factors was identified by the qRT-PCR and definitely, these results should be further validated by a series of molecular tools in the future.
Response:

We sincerely thank the reviewer for their valuable comments on our study, and we genuinely appreciate the positive and helpful summary. The reviewer's remarks have empowered us to refine our study. We believe that the revised manuscript has been significantly improved and is now suitable for publication in Plants.

Specific comments

Comment 2:

L19: Avoid using first-person writing throughout the manuscript.

Response:

We agree with the reviewer. As suggested, we have carefully removed personal pronouns and superfluous information.

Comment 3:

Keywords: These should be key terms but did not appear in the manuscript title.

Response:

We thank the reviewer for this comment and have revised the keywords, and it now reads:

“Moso bamboo; culm neck growth; anatomical and transcriptome analyses; primary thickening growth; lignin synthesis; nutrient transport; secondary metabolism; MYB.”

Comment 4:
Introduction: Add some background information about lignin biosynthesis in bamboo and the general roles of transcription factors that were investigated in this study, e.g. MYB family.

Response:

We thank the reviewer for this valuable comment. We have now included additional background information about lignin biosynthesis in bamboo and the general role of MYB family transcription factors. We believe this summarizes the background information effectively. It now reads: Please see line numbers 76-86.

“However, despite these developments, the transition mechanism regulating the rapid growth, particularly in relation to lignin biosynthesis, remains unknown. Lignin deposition in the secondary cell wall is a dynamic process [15], essential for providing structural support, rigidity, and resilience to bamboo [16]. Lignin biosynthesis involves a series of enzymatic reactions that ultimately deposit lignin precursors in the secondary cell wall. This process is regulated by transcription factors (TFs), such as the MYB family, which have emerged as key players in lignin biosynthesis, e.g., the MYB family members, such as MYB15, MYB46, MYB58, MYB63, CCoAOMT, etc. [17]. This suggests that the interplay between lignin biosynthesis and MYB TFs represents a crucial nexus in Moso bamboo development during the rapid growth [1].”

The newly added reference is copied below:

  1. Zhu, Y.; Huang, J.; Wang, K.; Wang, B.; Sun, S.; Lin, X.; Song, L.; Wu, A.; Li, H. Characterization of lignin structures in Phyllostachys edulis (Moso bamboo) at different ages. Polymers 2020, 12, 187, doi:10.3390/polym12010187.
  2. Li, L.; Yang, K.; Wang, S.; Lou, Y.; Zhu, C.; Gao, Z. Genome-wide analysis of laccase genes in Moso bamboo highlights PeLAC10 involved in lignin biosynthesis and in response to abiotic stresses. Plant Cell Rep 2020, 39, 751-763, doi:10.1007/s00299-020-02528-w.
  3. Xiao, R.; Zhang, C.; Guo, X.; Li, H.; Lu, H. MYB transcription factors and its regulation in secondary cell wall formation and lignin biosynthesis during xylem development. Int. J. Mol. Sci. 2021, 22, 3560, doi:10.3390/ijms22073560.

Comment 5:
Figure legends: should be enriched by adding information about abbreviations, e.g. the developmental stages, to make it easy to read figures individually and gene names should be in italics.

Response:

We thank the reviewer for this comment. We have ensured that the figure legends contain the necessary information about the abbreviations of the developmental stages. Now, all the figures are easy to read individually and stand on their own. Gene names are in italics.

Comment 6:
M&M: Add a section for statistics. Add information about how many replicates were conducted.

Response:

We thank the reviewer for pointing out this important point and have added a section for statistics. For transcriptome sequencing, there were five biological replicates for each stage (CNS3, CNS4, and CNS5). The statistics section now reads: Please see line numbers 519-525.

4.6. Statistical analysis and replicates

To ensure the reliability and robustness of our results, each stage of culm neck development (CNS3, CNS4, and CNS5) was represented by three biological replicates for anatomical analysis and five biological replicates for transcriptome sequencing. For qRT-PCR, three replicates were performed for each gene, providing additional validation of the transcriptome data. The data obtained from these replicates were subjected to appropriate statistical tests to determine the significance of differences in gene expression levels.”

Comment 7:

Discussion: The authors should discuss the general molecular mechanism more in detail on the angle change and comprehensively describe the present findings when compared to the literature.
Response:

We appreciate the reviewer's comments and have revised the discussion to provide a more detailed explanation of the angle change, particularly in the context of MYB family transcription factors involved in this process. We have also highlighted how our results contribute to the current understanding of Moso bamboo growth and culm neck development. It now reads: Please see line numbers 384-386 and 408-422.

“Notably, a genome-wide analysis pinpointed the involvement of key genes, including PALs, C4Hs, C3Hs, 4CLs, HCTs, CCRs, CCoAOMTs, CAD, F5Hs, and COMT, in the lignin biosynthesis process in Moso bamboo [35].

Like leaves affected by negative gravity, the culm neck grows upward against gravity. Leaf angle serves to orient leaves and other above-ground organs toward the sun and anchor them below ground [46,47]. Several genes are known to regulate leaf angles [46,47]. For instance, TFs such as OsBZR1 regulate OsIAA6 by interacting with the promoter of OsIAA6, which in turn controls leaf angles in rice by suppressing the auxin response factor, OsARF1 [48]. Similarly, OsARF19 regulates leaf angle by positively regulating OsGH35 and OsBRI1 and promoting cell division on the adaxial side of the lamina joint in rice [49]. Furthermore, OsMYB7, a member of the MYB TF family, plays a role in determining leaf angle during specific developmental stages in rice. When OsMYB7 is overexpressed, it results in the development of wide-angled leaves, while knockout mutants of osmyb7 display erect leaves. OsMYB7 achieves this effect by increasing the thickness of sclerenchyma cell walls, primarily through elevating cellulose content [50]. Interestingly, in the current study, MYB TF plays a role in the culm neck angle. These findings suggest that the molecular components responsible for the culm neck angle growth are similar to those found in leaf angle growth.”

The newly added reference is copied below:

  1. Peng, Z.; Lu, Y.; Li, L.; Zhao, Q.; Feng, Q.; Gao, Z.; Lu, H.; Hu, T.; Yao, N.; Liu, K. The draft genome of the fast-growing non-timber forest species moso bamboo (Phyllostachys heterocycla). Nat. Gene. 2013, 45, 456-461.
  2. Roychoudhry, S.; Kepinski, S. Shoot and root branch growth angle control—the wonderfulness of lateralness. Curr. Opin. Plant Biol. 2015, 23, 124-131, doi:https://doi.org/10.1016/j.pbi.2014.12.004.
  3. Mantilla-Perez, M.B.; Salas Fernandez, M.G. Differential manipulation of leaf angle throughout the canopy: current status and prospects. J. Exp. Bot. 2017, 68, 5699-5717, doi:10.1093/jxb/erx378.
  4. Xing, M.; Wang, W.; Fang, X.; Xue, H. Rice OsIAA6 interacts with OsARF1 and regulates leaf inclination. Crop J. 2022, 10, 1580-1588, doi:https://doi.org/10.1016/j.cj.2022.02.010.
  5. Zhang, S.; Wang, S.; Xu, Y.; Yu, C.; Shen, C.; Qian, Q.; Geisler, M.; Jiang de, A.; Qi, Y. The auxin response factor, OsARF19, controls rice leaf angles through positively regulating OsGH3-5 and OsBRI1. Plant Cell Environ. 2015, 38, 638-654, doi:10.1111/pce.12397.
  6. Kim, S.-H.; Yoon, J.; Kim, H.; Lee, S.-J.; Kim, T.; Kang, K.; Paek, N.-C. OsMYB7 determines leaf angle at the late developmental stage of lamina joints in rice. Front. Plant Sci. 2023, 14, 1167202, doi:10.3389/fpls.2023.1167202.

Comment 8:

Figure 11: I am curious about the related length of the culm neck when compared with the shoot buds. It seems too long in this figure.

Response:

We sincerely thank the reviewer for this valuable and very interesting comment. Unfortunately, we did not compare the length of the culm neck with the length of the shoot buds. However, the culm neck is smaller than the shoot bud. In our previous report [5], statistical analysis shows that the average length of the mature culm neck is about 31.2 mm, and the average diameter of the upper, middle, and lower parts is about 17.5 mm, 13.7 mm, and 12.1 mm, respectively.

  1. Sun, K.; Jiang, J.; Ding, Y.; Ramakrishnan, M.; Wei, Q. Morphological and anatomical analyses of moso bamboo culm necks. J. Nanjing For. Univ. (Nat. Sci. Ed.) 2021, 45, 40-46, doi:10.12302/j.issn.1000-2006.202103010.   

We have now redrawn the figure 11 to reflect your comment and copied the new figure below.   
